# Thyroid Hormone Changes Related to Growth Hormone Therapy in Growth Hormone Deficient Patients

**DOI:** 10.3390/jcm10225354

**Published:** 2021-11-17

**Authors:** Anna Małgorzata Kucharska, Ewelina Witkowska-Sędek, Małgorzata Rumińska, Beata Pyrżak

**Affiliations:** Department of Paediatrics and Endocrinology, Medical University of Warsaw, 02-091 Warsaw, Poland; ankucharska@wum.edu.pl (A.M.K.); malgorzata.ruminska@wum.edu.pl (M.R.); beata.pyrzak@wum.edu.pl (B.P.)

**Keywords:** hypothyroidism, thyroid hormones, growth hormone treatment, growth hormone deficiency

## Abstract

The alterations in thyroid function during recombinant human growth hormone (rhGH) treatment have been reported by many authors since this therapy became widely available for patients with growth hormone deficiency (GHD). Decrease of thyroxine level is the most frequent observation in patients treated with rhGH. This paper presents literature data describing changes in thyroid function related to rhGH therapy and a current explanation of mechanisms involved in this phenomenon. The effect of GH on the hypothalamic-pituitary-thyroid (HPT) axis is dependent on a multilevel regulation beginning from influence on the central axis, thyroid, and extra-thyroidal deiodinases activity as well as the impact on thyroid hormone receptors on the end. Changes in central and peripheral regulation could overlap during rhGH therapy, resulting in central hypothyroidism or an isolated slight deficiency of thyroxine. The regular monitoring of thyroid function is recommended in patients treated with rhGH and the decision of levothyroxine (L-thyroxine) supplementation should be made in the clinical context, taking into account thyroid hormone levels, as well as the chance for satisfactory growth improvement.

## 1. Introduction

The alterations in thyroid function during growth hormone (GH) therapy have been reported by many authors since the treatment with recombinant human GH (rhGH) became widely available for GH-deficient patients. A decrease in thyroxine level is the most frequent observation in patients treated with rhGH. This phenomenon was described as early as 1975 [1], and was later documented in studies in children as well as in adults treated with GH. The alterations in thyroid hormone balance during the rhGH treatment could have a divergent explanation and cause. It could be dependent on central hypothyroidism in the patients with hidden multihormonal hypopituitarism or could be directly dependent on the rhGH treatment and the influence of GH and insulin-like growth factor-1 (IGF-1) on thyroid hormone metabolism and balance. The character of the disorders is determined by the particular cause of GH deficiency. Nevertheless, it is still not clear whether and in whom the additional treatment with thyroid hormones is necessary.

The aim of the paper is to show the literature data describing changes in thyroid function related to GH treatment in patients with growth hormone deficiency (GHD) and to present a current explanation of mechanisms involved in this phenomenon.

## 2. Research of Literature Data

A systematic search for articles was conducted using MEDLINE and PubMed (until July 2021). The keywords used to search were “thyroid hormones” or “hypothyroidism” and “growth hormone treatment” or “rhGH therapy”. The initial number of papers was 559, and finally we found 30 papers that reported thyroid hormones changes in GHD patients (both children and adults) treated with rhGH (Table 1) [1,2,3,4,5,6,7,8,9,10,11,12,13,14,15,16,17,18,19,20,21,22,23,24,25,26,27,28,29,30]. A manual search of the references of the included papers was performed to identify any potential additional references.

Unfortunately, the results of the studies are inconsistent and do not strictly indicate the cause of the changes in thyroid hormone levels which had been reported. The main differences between the studies included a large heterogeneity of the patient groups, i.e., healthy individuals [31], non-GHD short children, non-GHD adults [32,33], and isolated idiopathic GHD or multiple pituitary hormonal deficiency (MPHD) in children and adults (Table 1). Furthermore, the rhGH doses administered in the patients and the duration of the observation periods were divergent.

In non-GHD subjects, alterations in thyroid function seemed to be less pronounced than in GHD patients [8,33,34,35,36,37]. In patients with GHD, both children and adults, the most typical finding was a significant increase of thyroxine (T4) to triiodothyronine (T3) conversion [5,7,8,10,11,12,17], but the elicitation of central hypothyroidism has also been reported in both children and adults, especially in those with malformations or organic lesions of the hypothalamic-pituitary region [13,14,19,27,38].

In the literature, there are only few studies evaluating changes in thyroid function during rhGH therapy in a homogenous group of children with isolated GHD [6,15,17,20,23,25,26,30]. Despite the reports differing in the number of studied children (from 19 to 117 individuals) and in the duration of the observation period (from 12 to 48 months), most of them showed a significant decrease in T4 and/or free thyroxine (fT4) levels without changes in thyroid-stimulating hormone (TSH) values. The two-year study by Seminara et al. [17] indicated that rhGH therapy led to significant changes in thyroid hormones during the first year of treatment, which may have resulted from alterations in the peripheral metabolism of thyroid hormones and seemed to be transitory in the second year of rhGH therapy [17]. A more recent study by Yao et al. [30] confirmed a significant reduction in fT4 levels during the two-year course of rhGH therapy but also showed a significant increase of TSH values at six months of therapy, which remained higher than the baseline to the end of the observation. The authors reported that none of the studied children developed hypothyroidism requiring levothyroxine (L-thyroxine) therapy [30]. The long-term study from our center [26], including the first four years of rhGH therapy, showed that the initiation of rhGH replacement led to a significant decrease in fT4 levels after the second year of treatment without significant changes in TSH values. Moreover, reduced fT4 levels compared to baseline values persisted in the next two years of rhGH therapy and were related to rhGH doses [26].

Most studies on the effects of rhGH therapy on thyroid function included mixed group of patients, both children and adults, with isolated GHD (IGHD) or MPHD, and their results are divergent (see Table 1). Changes in thyroid hormone levels due to alterations in their metabolism in peripheral tissues were reported in historical studies based on a small number of individuals in the 1970s and 1980s [5,7,10]. Wyatt et al. [11] in their study included the first year of rhGH therapy, which showed that the greatest decreases in T4, fT4 index and reverse triiodothyronine (rT3) levels and significant increases in triiodothyronine (T3) and the T3/T4 ratio were found after the first month of rhGH therapy. In subsequent months of therapy, a gradual return of serum thyroid hormones levels to the baseline values was observed. The authors concluded that there is a low risk of the development of clinically significant hypothyroidism in initially euthyroid GH-deficient children during growth promoting therapy and L-thyroxine supplementation is seldom necessary [11]. The study by Portes et al. [12] showed similar results to the above-mentioned analysis, as significant decreases in fT4 and rT3 levels coincided with significant increases in T3 levels independent of TSH values and resulted from increased peripheral conversion of T4 to T3. The authors suggested that rhGH replacement therapy does not lead to hypothyroidism, but only reveals previously unrecognized cases [12]. Smyczyńska et al. [22] assessed the effects of one-year rhGH therapy in a diverse group consisting of GHD children, children with GH neurosecretory dysfunction, and children with partial GH inactivity. They found a significant decrease in fT4 serum levels in the initial 3–6 months of rhGH administration without a significant increase in TSH values. They reported a relatively high number of patients (17 out of 67) who received L-thyroxine treatment due to a significant decrease in fT4 serum levels below the lower limit of the range or due to an increase in TSH levels above the normal range. Moreover, Smyczyńska et al. indicated that patients who became hypothyroid during rhGH treatment achieved a lower improvement in height velocity in the first year of therapy [22].

The unmasking of central hypothyroidism after the initiation of rhGH therapy has also been reported as a cause of the hypothalamic-pituitary-thyroid (HPT) axis alterations, especially in patients with severe GH deficit coincided with MPHD [13,14,19,27,38]. Porretti et al. [13], who studied the effects of six-month rhGH replacement therapy in adults with severe GHD, both isolated and in association with other multiple deficiencies, found that GH deficit, and consequently low IGF-1 levels, could mask central hypothyroidism in those patients. They reported that in 47% of the initially euthyroid individuals without any history of thyroid dysfunction, and in 18.3% of patients diagnosed earlier with central hypothyroidism with adequate baseline L-thyroxine substitution, fT4 levels significantly decreased below the reference values. The authors indicated the need to start L-thyroxine supplementation in previously untreated patients or to modify L-thyroxine dosage in patients diagnosed with central hypothyroidism before the initiation of rhGH therapy [13]. The one-year follow-up study by Giavoli et al. [14], based on a group of children with GHD, both idiopathic isolated GHD and GHD coinciding with MPHD, indicated that untreated GHD could mask the presence of central hypothyroidism in children with MPHD due to organic lesions of the hypothalamic-pituitary region [14]. Unfortunately, the limitation of that study is the limited number of patients (20 subjects with idiopathic isolated GHD and 6 subjects with MPHD) and the short duration of the observation. Agha et al. [19] studied the effects of rhGH therapy on more than 200 adults with severe GHD due to various hypothalamic-pituitary disorders. More than half of the study group were diagnosed with central hypothyroidism before the initiation of rhGH therapy and were treated with L-thyroxine, while the rest of the enrolled patients were considered euthyroid without thyroid dysfunction. The authors reported that in the untreated patients they observed a significant decrease in fT4 levels without changes in T3 and TSH serum values. Moreover, 36% of those patients became hypothyroid and adequate L-thyroxine substitution was administered. Similar, but less pronounced changes were observed in patients diagnosed with central hypothyroidism before the initiation of rhGH therapy, and 16% of them required an increase in L-thyroxine dosage. Agha et al. [19] indicated that low normal baseline fT4 serum levels in initially euthyroid GHD patients could predict the occurrence of MPHD during rhGH therapy. A nationwide retrospective cohort study by van Iersel et al. [27], including a large number of children with non-acquired GHD, revealed that central hypothyroidism occurred during rhGH therapy in more than 6% children with apparent isolated GHD. Most of them (75%) had congenital structural abnormalities of the pituitary. The authors indicated that GH-deficient children with fT4 levels in the lower range of reference values before the initiation of rhGH therapy are at risk of the occurrence of central hypothyroidism and require regular monitoring of their thyroid function. On the other hand, low baseline fT4 levels in children with congenital GHD are proposed as a predictor of structural pituitary lesions and diagnosis of MPHD [27]. An interesting idea was postulated recently by Ebuchi et al. [29], who reported that a thyrotropin-releasing hormone (TRH) stimulation test, conducted before the initiation of growth promoting treatment, could help to identify individuals who are at risk of central hypothyroidism due to rhGH therapy and could require L-thyroxine replacement therapy. Nevertheless, this method of prediction seems to be relatively expensive and not widely available.

## 3. Mechanisms Explaining the GH/IGF-1 Effect on the HPT Axis

The mutual interplay between the GH/IGF-1 axis and the HPT axis is variable. Biological evidence of close relationships between the two axes are mainly based on clinical observations in GH-deficient patients treated with rhGH at physiological doses and also in opposite circumstances—in acromegalic patients with pathological GH excess, in whom the thyroid gland is often affected. In patients with supraphysiological level of GH, goiter is reported in 55–87% [39,40] and thyroid cancer in 8.7–10.6% [41,42,43].

### 3.1. The HPT Axis Suppression in the Patients Treated with rhGH

Jørgensen et al. [8] postulated the direct inhibitory effect of rhGH to pituitary circadian secretion of TSH levels. In line with this data is the lack of TSH increase associated with the decrease of fT4 observed in the majority of GHD patients treated with rhGH (see Table 1). These data are also supported by the study of Roelfsema et al. [44], wherein there were untreated patients with acromegaly who had reduced basal and pulsatile TSH secretion with lower amplitude. The authors explained this phenomenon by increased (by GH) intracellular conversion of T4 into T3 in folliculo-stellate cells of the pituitary, which via biofeedback inhibits the synthesis of TSH beta-subunit and the secretion of mature TSH [44,45]. This mechanism seems to be very attractive as the explanation of GH’s effect on pituitary; however, it should be taken into account that the levels of GH in acromegalic patients are much higher in comparison to GH substitution in GHD patients, in whom there should be guaranteed physiological GH/IGF-1 levels.

Studies in GHD children suggest that the absence of the nocturnal TSH increase could predict the development of hypothyroidism during rhGH replacement [46]. Additionally, in response to increased GH concentration, enhanced somatostatin (SST) secretion is expected, which directly suppresses TSH release via SST receptors (SST2 and SST5). Moreover, diminished secretagogues release, especially ghrelin, might also decrease the TSH secretion by blunting of the TRH effect [47,48]. Decreased ghrelin was reported in adult GH-deficient patients in a response to rhGH [47] treatment as well as in children [48]. Another neuroregulatory factor involved in the HPT axis and GH connections is the leptin pathway. Decreased leptin level was reported in GHD children in association with rhGH treatment. This could result from a direct effect of GH; nevertheless, the regulation of leptin could also play some role in the TRH/TSH suppression in patients treated with rhGH considering the changes of body composition resulting in a reduction of fat tissue, which could explain the leptin decrease [49].

### 3.2. Peripheral Thyroid Hormone Metabolism Alterations during rhGH Therapy

The most frequent alteration of thyroid hormone balance in patients under rhGH treatment is decreased fT4 and rT3 with increased fT3. These changes are explained by the influence of GH on the activity of deiodinases (DIOs: DIO1, DIO2, and DIO3). Glynn et al. [24] investigated the activity in subcutaneous adipose tissue in 20 adult GHD males treated with rhGH. In their study, there was a correlation between the decrease of circulatory fT4 and increased fT3 levels with the dose of rhGH, but the activity of the DIO2 enzyme in subcutaneous fat declined and DIO1 and DIO3 activities were not changed during rhGH therapy [24,50]. Nevertheless, the effect on peripheral thyroid hormone balance could be dependent on liver deiodinases activity, which are considered as a main source of circulating T4 to T3 conversion and the influence of GH on DIO could have a divergent effect in subcutaneous tissue than in the liver. The liver deiodinases are sensitive to direct GH stimulation. Hepatocytes have no functional receptors for IGF-1, whereas GH receptors are expressed in these cells and can stimulate hepatic deiodinases [51]. In the study by Yamauchi et al. [28], it was established that treatment with rhGH increases serum fT3 and decreases serum fT4 levels, which is associated with increased DIO2 activity in the thyroid cell line culture. Previous reports in animals showed that DIOs activity is directly influenced by GH, but the distribution of this enzymes varies in different animal species [52,53]. In humans, DIO2 is highly expressed in thyroid cells [54]. In the study by Yamauchi et al. [28], in vitro administration of GH significantly increased DIO2 expression at the mRNA level and protein activity in the culture of human thyroid HTC/C3 cells. Glynn et al. [50] proposed an explanation of the rise in T3 by its reduced clearance dependent on decreased DIO3 activity. Nevertheless, the depression of DIO3 was not confirmed in other studies of the effect of rhGH on DIOs [28]. During rhGH therapy, the effect on DIOs activity seems to be directly dependent on GH and not mediated by IGF-1. This idea is supported by studies by Hussain et al. [55], who reported that serum fT3 levels increase much more after rhGH than after IGF-1 administration. A similar finding was described during IGF-1 treatment in patients with Laron-type dwarfism [56]. However, the expression of IGF-1 receptors was well documented in human thyroid cells [57,58] and the proliferative effect of IGF-1 was proved in vitro in thyroid follicular cell cultures [59], but this relationship impacts thyroid size and nodular changes and not the hormonal function.

### 3.3. GH Impact on Thyroid Hormone Receptors

An additional mechanism which could be involved in thyroid hormone balance disorders after rhGH therapy is its impact on thyroid hormones receptors. The presence of thyroid hormone receptors (TRs) in growth plates is well known [60]. Susperreguy et al. [61] reported that the individual growth response in patients with idiopathic short stature treated with rhGH correlated positively with the change in the TRα mRNA level and negatively with the TRβ mRNA level. The design of the study was based on the evaluation of mRNA of TRs in peripheral blood mononuclear cells; however; there was a parallel evaluation of serum sex hormone binding globulin which, reflects the effect of thyroid hormones on the liver, and also osteocalcin and b-cross laps, which are serum biochemical markers reflecting thyroid hormone action at the bone tissue. The authors reported that the higher increase in TRα1 and TRα2 mRNAs levels after 12 months of GH treatment was positively correlated to the gain of growth velocity and bone markers. Moreover, they observed a better gain in growth velocity in response to GH therapy in the individuals who had the higher increase of TRα1 mRNA level [61]. This finding is in line with the main role of TRα1 in bone longitudinal growth and suggests a possible direct influence of GH on thyroid hormone receptors in the growth plate.

Considering the above, we can summarize that the effect of GH on thyroid hormones is dependent on a multilevel regulation involved influence on the central axis, thyroid, and extra-thyroidal deiodinases activity and thyroid hormone receptors (Figure 1). Numerous elements of this puzzle have been well-documented in human as well as in animal models [52,53] and in cell line cultures [55,57,58,59]. Changes in central and peripheral regulation could overlap during rhGH therapy, resulting in central hypothyroidism in some patients or the isolated slight deficiency of thyroxine in others.

## 4. Therapeutic Implications of the rhGH Influence on Thyroid Function

The unsolved issue is the indication for the L-thyroxine substitution in the patients treated with rhGH. Only a few patients develop overt hypothyroidism, but there is clinical evidence of the positive impact of higher thyroxine levels on growth velocity in children treated with rhGH [62]. Thyroxine enhances the effect of GH, directly stimulating the growth hormone receptor (GHR)/Janus kinase 2 (JAK2)/signal transducer and the activator of the transcription 5 (STAT5) pathway [62], and therefore improves the growth effect during rhGH treatment. The regular monitoring of thyroid function is recommended in patients treated with rhGH [63]. On the other hand, there are some different opinions. Giavoli et al. [14], based on a group of children with GHD, both idiopathic isolated GHD and MPHD, indicated that untreated GHD could mask the presence of central hypothyroidism in children with MPHD due to organic lesions of the hypothalamic-pituitary region. However, the authors concluded that there is no need to monitor thyroid function in children with idiopathic isolated GHD because of no risk of central hypothyroidism in that group of patients [14]. Similarly, de Kort et al. [34], including more than 250 short non-GH-deficient prepubertal children born small-for-gestational age, showed that rhGH therapy led to a significant decrease in fT4 levels, but that effect neither coincided with an increase in TSH values nor affected the growth response to rhGH therapy. They also concluded that regular monitoring of thyroid hormone levels during rhGH therapy is not warranted in such children [34]. On the other hand, a recently published expert opinion is that “there are no absolute criteria enabling exclusion of children without any risk of progress to combined pituitary hormone deficiency” [64], and lifelong monitoring of the pituitary gland function is recommended, especially in patients with organic GHD [64].

## 5. Conclusions

In conclusion, we can summarize that rhGH treatment influenced the HPT axis on the central as well as on the peripheral level and changes could appear at any time during therapy. The group with a high risk of hypothyroidism are patients with MPHD or organic GHD, but it should also be considered in patients with a deteriorated growth effect during rhGH therapy. The decision of L-thyroxine supplementation should be made in the clinical context, taking into account thyroid hormone levels as well as a satisfactory growth effect during rhGH treatment.

## Figures and Tables

**Figure 1 jcm-10-05354-f001:**
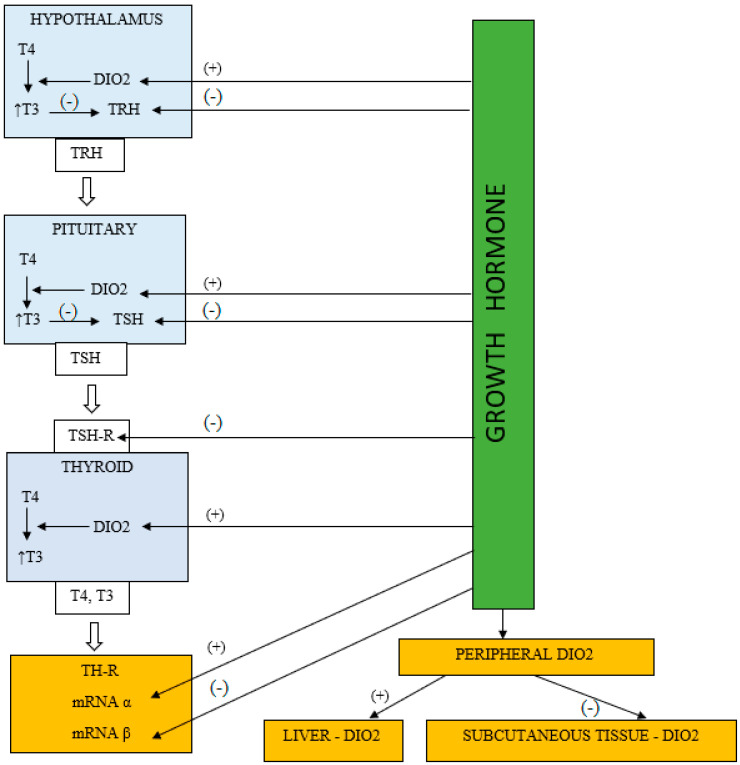
Hypothetic pathways of direct modulation of the hypothalamic-pituitary-thyroid receptors axis by growth hormone. TH-R—thyroid hormone receptor, DIO2—deiodinase 2, fT4—free thyroxine, fT3—free triiodothyronine, TSH—thyroid-stimulating hormone, TRH—thyrotropin-releasing hormone.

**Table 1 jcm-10-05354-t001:** Characteristics of the studies evaluating effects of rhGH therapy on thyroid function in GHD patients.

Authors	Year	Diagnosis	No of Patients	Study Population	T4 and/or fT4	T3 and/or fT3	rT3	TSH
Porter et al. [2]	1973	IGHD/MPHD	6	Children	↔	n.a.	n.a.	↓
Lippe et al. [1]	1975	IGHD/MPHD	6	Children	↓	↓	n.a.	↓
Cacciari et al. [3]	1979	IGHD/MPHD	24	Children	↔	↔	n.a.	↔
Demura et al. [4]	1980	IGHD/MPHD	26	Children	↓	↔	n.a.	↔
Jørgensen et al. [5]	1989	IGHD/MPHD	22	Adults	↓	↑	↓	↔
Pirazolli et al. [6]	1992	IGHD	57	Children	↓	↑	n.a.	↔
Rezvani et al. [7]	1992	IGHD/MPHD	7	Children	↔	↑	↓	↔
Jørgensen et al. [8]	1994	IGHD/MPHD	10	Adults	↔	↑	↓	↓
Monson et al. [9]	1994	MPHD	21	Adults	↓/↔	↑/↔	n.a.	n.a.
Sato et al. [10]	1977	IGHD/MPHD	8	Children	↓	↑/↔	n.a.	↑/↔
Wyatt et al. [11]	1998	IGHD/MPHD	15	Children	↓	↑	↓	↔
Portes et al. [12]	2000	IGHD/MPHD	20	Children	↓	↑	↓	↔
Porretti et al. [13]	2002	IGHD/MPHD	66	Adults	↓	↔	↓	↔
Giavoli et al. [14]	2003	IGHD/MPHD	26	Children	↓	↔	n.a.	↔
Kalina-Faska et al. [15]	2004	IGHD	32	Children	n.a.	n.a.	n.a.	↓
Hubina et al. [16]	2004	MPHD	112	Adults	↔	↑/↔	n.a.	↔
Seminara et al. [17]	2005	IGHD	19	Children	↓	↑	n.a.	↔
Martins et al. [18]	2007	IGHD/MPHD	30	Children/adults	↓	↑	n.a.	n.a.
Agha et al. [19]	2007	MPHD	243	Adults	↓	↔	n.a.	↔
Moayeri et al. [20]	2008	IGHD	21	Children	↓	↔	n.a.	↔
Losa et al. [21]	2008	IGHD/MPHD	49	Adults	↓	↔	n.a.	↔
Smyczynska et al. [22]	2010	IGHD/NSD/inactGH	75	Children	↓	n.a.	n.a.	↔
Ciresi et al. [23]	2014	IGHD	105	Children	↔	↑	n.a.	↔
Glynn et al. [24]	2017	IGHD/MPHD	20	Adults	↓	↑	↓	↔
Keskin et al. [25]	2017	IGHD	29	Children	↓	2194	n.a.	↓
Witkowska-Sędek et al. [26]	2018	IGHD	117	Children	↓	n.a.	n.a.	↔
Van Iersel et al. [27]	2018	IGHD/MPHD	456	Children	↓	n.a.	n.a.	↔
Yamauchi et al. [28]	2018	MPHD	20	Adults	↔	↑	n.a.	↓
Ebuchi et al. [29]	2020	IGHD/SGA	203	Children	↓GHD/↑SGA	n.a.	n.a.	↓
Yao et al. [30]	2021	IGHD	55	Children	↓	n.a.	n.a.	↑

rhGH—recombinant human growth hormone, IGHD—isolated growth hormone deficiency, MPHD—multiple pituitary hormones deficiency, NSD—neurosecretory dysfunction, inactGH—partial GH inactivity, SGA—small for gestational age, T4—total thyroxine, fT4—free thyroxine, T3—total triiodothyronine, fT3—free triiodothyronine, rT3—reverse triiodothyronine, TSH—thyroid stimulating hormone, ↔—no changes, ↑—increased, ↓—decreased, n.a.—not available.

## Data Availability

The study is based on published papers available in Pubmed and Medline and did not report any additional data.

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
