# Peer review of "Thyroid Hormone Changes Related to Growth Hormone Therapy in Growth Hormone Deficient Patients"

_jcm, 2021, doi:10.3390/jcm10225354_

Round 1

Reviewer 1 Report

  1. the title section – Is the word “depression” used correctly in this context?
  2. line 10 – rhGH – please explain the abbreviation (always explain them when they appear in the text for the first time)
  3. line 13 – HTP – please explain the abbreviation – the way you do it in line 117: “hypothalamic-pituitary-thyroid (HPT) axis” should be used in the abstract.
  4. abstract section – proofreading will be necessary, as there are already some mistakes in the abstract section
  5. Figure 1 – the figure is not very clear to me – can you add some explanation to the figure or maybe redo it so that it is more clear to an average reader who’s not a specialist in GH therapy?
  6. The conclusion section – this section could be more detailed, the conclusions presented by the authors are quite obvious – tests always have to be carried out, the dose and the chances od success also have to be determined. The study presents the findings from other authors in a comprehensive and interesting way, but lacks a more detailed conclusions section or perhaps a short discussion section.

Author Response

Thank You for your thoroughly analysis, your precious advices we hope significantly  improve the value of the paper.

  1. We changed the title of paper, we agree that “depression” in that context is not appropriate, because the aim of our study was to describe in complexity of growth hormone effect on thyroid hormones balance and also the HPT axis.   
  2. line 10 - we inserted the explanation of the abbreviations everywhere when they appear in the text for the first time
  3. We changed the Figure 1 , in which we show the possible pathways of direct effect of GH on central and peripheral regulation of thyroid hormone balance.
  4. We changed the conclusion section and add some short discussion at the end of the part about the indications to the treatment.  

Reviewer 2 Report

The recommendations are in the attached file

Author Response

Dear Reviewer,

thank you very much for your precious comments. We appreciate for your advices and we included all of it in the reconstruction of our paper, what ,we hope, helped to improve the value of our paper.

We changed the title of the paper which  more precisely describe the issue.

We explained each abbreviation by its first appearance in the text.

We added the key words, which were used in the systemic search in Medline and Pubmed,

We included the discussion about thyreotropin-releasing hormone stimulation test reported by Ebuchi Y et al.,  and other references in the text.  Considering the contribution of ghrelin, or leptin in the relationships between GH axis and HPT axis we included this information in the text at the end of part 3.1. The influence of this peptides on HTP axis is of course important, but during rhGH treatment it is difficult to differentiate the consequences of rhGH therapy and implications of changes of body composition in treated patients. The changes of ghrelin, leptin or other adipokines in the response to rhGH therapy modulate physiological effects of this peptides. In our paper we focused the main attention on direct effects of GH on HPT axis, and we only mentioned the influence of additional fators.

THE STRUCTURE OF THE REVIEW The part 2.

We reconstructed the part 2, but the better unification of study groups was limited because of source papers, in which were reported observations in the patients with different kind of GH deficiency. Nevertheless we tried to make it.

In part 3.1. we mentioned some data about HPT axis observation in patients with acromegaly and we explain of course that in this condition the levels of GH are much higher, than doses in GH treatment. The idea of this citation was that some effects of GH could be more evident in the patients under supraphysiological levels of GH.

In part 3.3. in the fragment discussing thyroid hormone receptors we added discussion about the significance of the results in comparison to bone markers in analyzed group of patients, what allowed to conclude that alterations  mRNA of THR in PBMC could reflect the similar changes in bones. 

Thank you once more for your precious remarks.

Round 2

Reviewer 1 Report

All my commentaries have been satisfactorily replied to. I think this article can be now considered for publication

Author Response

Thank You very much for your revision.

Reviewer 2 Report

  1. The title of the article sounds well. The additional explanation “Possible  Mechanisms based on Literature Review” is not mandatory, while the article is already defined as review

The word “alterations” seems rarely used in the connection with the hormone measurements – maybe the word “changes” would be more appropriate

  1. The expression in the first sentence of conclusions should be more general in the description of the changes – the possible correction is advised (line 781)

In conclusion, we can summarize that rhGH treatment influences the HPT axis on  central as well as on peripheral level and ...... thyroid hormone changes could be measured at any time during the therapy

Author Response

Thank you very much for the second revision.

Accordingly to your advices we changed the title for: "

Thyroid Hormones Changes Related to Growth Hormone Therapy in Growth Hormone Deficient Patients."

and changed the sentence in conclusions like that:

"In conclusion, we can summarize that rhGH treatment influenced the HPT axis on central as well as on peripheral level and changes could appear at every time of therapy. 

previously was: " and deficiency of thyroxine could appear at every time of therapy" .